# Seroprevalence of Diphtheria and Tetanus Immunoglobulin G among the General Health Population in Guangzhou, China

**DOI:** 10.3390/vaccines12040381

**Published:** 2024-04-04

**Authors:** Qing He, Yejian Wu, Shuiping Hou, Lei Luo, Zhoubin Zhang

**Affiliations:** 1Guangzhou Center for Disease Control and Prevention, Guangzhou 510440, China; heqingnzzz@foxmail.com (Q.H.); wuyejian@163.com (Y.W.); gzcdc367@163.com (S.H.); 2Institute of Public Health, Guangzhou Medical University & Guangzhou Center for Disease Control and Prevention, Guangzhou 511436, China

**Keywords:** diphtheria, tetanus, seroepidemiology, immunization schedule

## Abstract

A seroepidemiological study was conducted in 2018 to assess diphtheria and tetanus antibodies in Guangzhou, China. Diphtheria and tetanus antibody concentrations were measured with an enzyme-linked immunosorbent assay. A total of 715 subjects were enrolled in the study. The overall diphtheria and tetanus toxoid IgG-specific antibody levels were 0.126 IU/mL (95% CI: 0.115, 0.137) and 0.210 IU/mL (95% CI: 0.185, 0.240), respectively; the overall positivity rate was 61.82% (95% CI: 58.14, 65.39) and 71.61% (95% CI: 68.3, 74.92), respectively. The diphtheria and tetanus antibody concentration was decreased by age and increased by doses. The geometric mean concentrations and positivity rate of diphtheria and tetanus antibodies were lowest and below the essential protection level in people over 14 years of age. Compared to children and adolescents, middle-aged people and the aged are at much higher risk of infection with Corynebacterium diphtheriae and Clostridium tetani. The current diphtheria and tetanus immunization schedule does not provide persistent protection after childhood. There is an urgent need to adjust the current immunization schedule.

## 1. Introduction

Respiratory diphtheria is an acute, contagious infectious illness caused by toxigenic strains of Corynebacterium diphtheriae that can lead to difficulty breathing, heart failure, paralysis, or death [1]. Tetanus is an acute, often fatal, disease caused by the toxin of the bacterium Clostridium tetani and is characterized by muscle spasms and autonomic nervous system dysfunction [2].

Diphtheria was well-controlled in China due to the effective vaccine. The most recent diphtheria case nationwide was reported in 2006, down from the maximum of 70.7 thousand cases (11.1 per 100,000 people) recorded from 1950–1965 [3]. However, over the last few years, patients carried non-pathogenic, non-toxigenic corynebacterium diphtheria has been increasingly reported in China, 2018 in Dongyang [4], 2022 in Zhuhai [5], 2019 [6], 2023 and 2024 in Guangzhou. The research found that the ß-corynebacteriophage can infect non-toxigenic strains, which leads to transformation to a toxigenic strain and production of the diphtheria toxin [7]. Worse yet, diphtheria outbreaks continued to be reported in developing countries, South Africa [8], Yemen [9], Venezuela [10], India [11], and Malaysia [12]. Developed countries that have eradicated diphtheria have seen a reemergence of the disease, like Switzerland [13] and Germany [14]. In 2020, Xiamen, China, reported an imported cutaneous diphtheria [15]. These could pose threats to China.

The elimination of maternal and neonatal tetanus was confirmed in China in 2012. The reported neonatal tetanus rate decreased from 1079 cases/year in 2008 to 15 cases/year in 2022 [16,17]. However, the success was mainly due to improvements in the medical environment and increased hospital delivery rates rather than immunization programs. [18]. Since 2015, no neonatal tetanus case has been reported in Guangzhou, China (unpublished data). However, non-neonatal tetanus, which means tetanus cases occurring after the neonatal period (aged > 28 days), is not a national notifiable disease in China. There is a lack of systematic tetanus epidemiological surveillance and reporting systems among non-neonatal patients in China. Post-traumatic tetanus is prevalent in rural areas, with a high misdiagnosis rate and missed diagnosis rate [18]. A retrospective study of tetanus cases in the Anhui Province of China from 2013 to 2022 showed that 97.85% of tetanus patients were over 18 years old [19]. Medical records of the hospital in Guiling, China, in 2015–2017 revealed that 94.20% of the tetanus patients were over 40 years old, and the case fatality rate was up to 37.68%, with a morbidity of 0.43/100, 000 [20]. The reported tetanus morbidity may be underestimated, true tetanus incidence is uncertain.

There are two categories, four types of diphtheria and tetanus toxoid-containing vaccines (DTCV) used in China. Category 1 vaccines [21] are provided free of charge to residents, are mandatory for children, and are manufactured in China; they include diphtheria and tetanus toxoids with acellular pertussis vaccine (DTaP) and absorbed tetanus and reduced diphtheria combined vaccine (DT). Category 2 vaccines [21] are considered and are paid for by the vaccinee or their guardian, including diphtheria, tetanus, acellular pertussis and Haemophilus influenza type b combined vaccine (DTaP-Hib), diphtheria, tetanus, acellular pertussis, poliomyelitis and Haemophilus influenza type b combined vaccine (DTaP-IPV/Hib).

Combined diphtheria, tetanus, and whole-cell pertussis vaccine (DTwP) were first licensed in the 1940s, and monovalent diphtheria vaccines started to be used in China in 1953 [22]. In the 1970s, DTwP began to replace the monovalent vaccines in China. The incidence of diphtheria has been significantly reduced since China launched the planned regular vaccination program with DTaP in 1978. DTwP replaced DTaP gradually due to adverse events since 2007. DTaP and DTwP were used from 2007 to 2012, but since 2013, only DTaP has been administered [3]. China’s latest diphtheria and tetanus immunization schedule is three doses of DTaP at 3, 4, and 5 months, a booster dose at 18 months, and one dose of DT at six years of age (Figure 1). Five doses of DTCV in children are received in the Chinese national immunization schedule. Category 1 vaccines in the national immunization schedule are allowed for replacement with Category 2 vaccines.

It should be noted that there is no DTP or DT booster during pregnancy or booster vaccinations against diphtheria and tetanus in teens and adults in China. Also, DT dosage forms used in China only apply to children, not adults.

The level of antibodies against infectious diseases, as measured by serosurveillance, results from vaccination or previous infection history and can be used as an indicator of the immune ability of individuals to diseases and the effectiveness of the target vaccine in the vaccinated population. Also, monitoring the vaccine-preventable disease immunity levels is the responsibility of the Centers for Disease Control and Prevention, as stipulated by the new China vaccine administration law. A seroepidemiological study has been conducted annually to assess the level of humoral immunity against communicable diseases and to monitor the population’s vaccine-preventable disease immunity levels in healthy people in Guangzhou, China.

According to the actual situations, we tested immunoglobulin G (IgG) against diphtheria toxoid and tetanus toxoid to determine the current individual immune status in healthy people, to find out the vulnerable group to diphtheria and tetanus, to evaluate the durability of diphtheria and tetanus antibody persistence after five doses DTCV immunization schedule, and to provide evidence for adjusting the current diphtheria and tetanus immunization schedule in China. We supposed that diphtheria and tetanus antibody levels waned after vaccination and were lowest in adults. 

## 2. Materials and Methods

### 2.1. Study Design

Annual monitoring of population immunity levels in healthy people based on the community was conducted in Guangzhou. Three districts of Guangzhou were selected using multi-stage stratified sampling methods based on geographical location (central urban area, suburban area district, outer suburb area district). One or more communities were chosen from each district by simple random sampling. Subjects were recruited from the above communities. Subjects’ venous blood and demographic information (age, sex, address, household registration, date of birth, and date of sampling) were collected by community staff from July to December 2018.

### 2.2. Sample Size

Based on the calculation formula of a cross-sectional study, the minimum number of subjects was calculated to be 442, with the expected diphtheria antibody seroprevalence of 50%, a precision of 0.05, an allowable error of 0.05 and a 15% non-response rate. Each district in each age group (<1, 1–2, 3–4, 5–6, 7–10, 11–14, 15–19, 20+ years old) recruited at least 19 subjects.

### 2.3. Basic Information on the Study Site

Guangzhou is located in eastern Asia, the largest trading city in southern China, and the political, economic, and cultural center of Guangdong Province. This city has a registered population of over 9.28 million people and a floating population (defined as living in Guangzhou and a nonlocal census register) of over 5.63 million people in 2018. Guangzhou is vital in connecting mainland China with other Southeast Asian countries.

The three selected districts were Yuexiu (central urban area), Baiyun (suburban area), and Zengcheng (outer suburb area district). The population sizes of Yuexiu, Baiyun, and Zengcheng were 1.18, 2.71, and 1.22 million in 2018, respectively.

### 2.4. Inclusion and Exclusion Criteria of Participants

Inclusion criteria: (1) healthy people without fever (axillary temperature ≤ 37.1 °C), without acute disease, and symptoms of respiratory infection on the day of blood collection; (2) people who had resided for at least three months in their community in Guangzhou; (3) subjects or their guardians who were willing to participate in this surveillance voluntarily.

Exclusion criteria: all individuals receiving any medical care for any condition.

### 2.5. Serologic Evaluations

Blood samples were centrifuged at 3500× *g* for 15 min, and sera were transferred into 2 mL cryotubes under sterile conditions. Serum samples were stored at −80 °C at the Guangzhou Center for Disease Control and Prevention (GZCDC). Before testing, sera were let stand for 1 h and processed when they reached room temperature. An enzyme-linked immunosorbent assay (ELISA; Virion/Serion, Würzburg, Germany) was used to measure anti-diphtheria toxoid IgG titer or anti-tetanus toxin titer quantitatively. The assay was performed by GZCDC from February to April 2019, according to the manufacturer’s instructions, and optical density was measured at a wavelength of 450 nm using a MicroplateReader (Tecan Sunrise, Männedorf, Switzerland). They were expressed in international units per milliliter (IU/mL). 

According to the manufacturer’s instructions, the antibodies against diphtheria toxin were divided into six groups: <0.01 IU/mL (undetectable/no protection), 0.01–<0.1 IU/mL (no protection), 0.1–1.0 IU/mL (short term protection/basic protection) and >1.0 IU/mL (long term protection). The antibodies against tetanus toxin were divided into six groups: <0.01 IU/mL (no immunity), 0.01–<0.1 IU/mL (immune protection is not ensured), 0.1–1.0 IU/mL (adequate immune protection), and >1.0 IU/mL (long term protection). The analysis was facilitated by defining seroprotection (concentration likely to predict protection) with cut-off values of 0.1 IU/mL for both antibodies against diphtheria and antibodies against tetanus.

### 2.6. Groupings and Definition

The age group was divided into three groups: 0–1 year old, 2–6 years old, 7–14 years old, and 15+ years old.

Household registration was divided into two groups based on whether or not their census register was in Guangzhou, including the local and nonlocal census registries. 

Interval since the last vaccination was defined as an interval (year) between the last DTCV vaccination date and the date of blood collection.

DTCV dose record was defined as the DTCV doses children have received and recorded in the Guangdong Vaccine Circulation and Vaccination Management Information system. Diphtheria toxoid-containing vaccines and tetanus toxoid-containing vaccines are combined in one vaccine (DTaP, DT, DTaP-Hib DTaP-IPV/Hib), the records of diphtheria toxoid-containing vaccines and tetanus toxoid-containing vaccines are the same.

The DTCV vaccination date was defined as the date of the last DTCV dose children received, and it was recorded in the Guangdong Vaccine Circulation and Vaccination Management Information system.

### 2.7. Vaccination History Collected

DTCV vaccination history (doses, vaccination date) was collected from the Guangdong Vaccine Circulation and Vaccination Management Information system by matching participants’ names, sex, date of birth, and address. 

The Guangdong Vaccine Circulation and Vaccination Management Information System was an electronic network system for vaccine management, circulation, and delivery in Guangdong. Each child will be create a file with their personal information since they were born in Guangdong. Information about each dose of vaccine children received will be recorded in their file in the system by the nurse at the time of vaccination. The record information includes the vaccine’s name, manufacturer, doses, batch number, injection site, immunization date, and vaccination institution. When the nonlocal census registers children who receive vaccines in Guangzhou, the nurse will extract their vaccination files to record their vaccination information. Entering the vaccination information is stringently enforced in children who received vaccines but not in adults. Given the year of development of the information system and the risk of recall bias, we only collected vaccination information for children under 15 years old. 

### 2.8. Statistical Analysis

Data processing and analysis were performed using R version 4.3.0, along with Storm Statistical Platform 1.0 (www.medsta.cn/software, accessed on 3 March 2023). The diphtheria or tetanus antibody concentration was recorded as geometric mean concentrations (GMCs), and logarithmic analysis was performed. Two-sample *t*-test (or Wilcoxon two-sample test), one-way ANOVA (or Kurskal–Wallis test), or χ^2^ test were used to compare the difference of diphtheria antibody concentration or demographic information between groups within the same doses. ANOVA trend test was used to test the trend of antibodies by age group and DTCV doses. Multivariable regression analysis with a stepwise method examined the association between log-transformed diphtheria or tetanus antibody concentration and groups. The strategy of stepwise regression is stepwise selection (or sequential replacement), which starts with no predictors and then sequentially adds the most contributive predictors (*p* ≤ 0.05). After adding each new variable, remove any variables that no longer provide an improvement in the model fit (as measured by AIC).

When analyzed, the missing data on DCV dose records and vaccination dates are ignored.

## 3. Results

### 3.1. Eleven Communities from Three Districts Were Selected in the 2018 Surveillance of Population Immunity Levels

A total of 715 subjects were enrolled in the study, ranging from newborn to 71 years old, with a median age of 5 and an average age of 9.26. The ratio of male to female was 1:0.98. The ratio of local residents to nonlocal residents was 1.45:1. About 80% (581/715) of the subjects were aged range between 0–14 years old, with 552 (95.01%, 552/581) having definite DTCV dose records, 549 (94.49%, 549/581) having definite DTCV vaccination date. 

### 3.2. Diphtheria Antibody Levels

The overall diphtheria toxoid IgG-specific antibody levels were 0.126 IU/mL (95% CI: 0.115, 0.137), and the age group showed a statistical significance decrease from 0.125 IU/mL (0 y) to 0.048 IU/mL (21 y+), were lowest in people over 14 years old (0.032 IU/mL). Males had higher diphtheria toxoid IgG-specific antibody levels than females. The central urban area had higher diphtheria toxoid IgG-specific antibody levels than the suburban and outer suburban areas. The diphtheria toxoid IgG-specific antibody levels showed a statistically significant increase in the DCV doses of subjects who received from 0 doses (0.040 IU/mL) to 5 doses (0.193 IU/mL) (*p* < 0.001) (Table 1).

### 3.3. Tetanus Antibody Levels

The overall tetanus toxoid IgG-specific antibody levels were 0.210 IU/mL (95% CI: 0.185, 0.240), and the age group showed a statistical significance decrease from 0.169 IU/mL (0 y) to 0.023 IU/mL (21 y+). They were lowest in people over 14 years old (0.017 IU/mL). Females had higher diphtheria toxoid IgG-specific antibody levels than male and nonlocal people. The central urban area had higher diphtheria toxoid IgG-specific antibody levels than the suburban and outer suburban area. The tetanus toxoid IgG-specific antibody levels increased by the DCV doses of subjects who received from 0 doses (0.015 IU/mL) to 5 doses (0.282 IU/mL) (*p* < 0.001) (Table 1).

The tetanus toxoid IgG-specific antibody levels for women of childbearing age were 0.013 (95% CI: 0.008, 0.021), significantly lower than overall subjects.

For diphtheria and tetanus toxoid IgG-specific antibodies, the differences between groups by age group, sex, area, and doses were all statistically significant (*p* < 0.05), except for household.

### 3.4. Diphtheria Antibody Positivity Rate

The overall positivity rate of diphtheria antibodies was 61.82% (95% CI: 58.14, 65.39), which increased from 0 to 7–14 y and then decreased rapidly in people over 14 y old (15.67% (95% CI: 9.44, 21.91)) with statistical significance (*p* < 0.05, Table 1). Males had a higher diphtheria toxoid IgG-specific antibody positivity rate than females. The central urban area had a higher diphtheria toxoid IgG-specific antibody positivity rate than the suburban and outer suburban areas. The diphtheria toxoid IgG-specific antibody positivity rate showed a statistically significant increase by the DCV doses of subjects who received from 0 doses (21.62%) to 5 doses (79.17%) (*p* < 0.001). 

### 3.5. Tetanus Antibody Positivity Rate

The overall positivity rate of tetanus antibody was 71.61% (95% CI: 68.3, 74.92), highest in children aged 1 y and lowest in people over 14 y. Males had a higher tetanus toxoid IgG-specific antibody positivity rate than females. Central urban and suburban areas had a higher tetanus toxoid IgG-specific antibody positivity rate in the outer suburb area. The tetanus toxoid IgG-specific antibody positivity rate showed a statistically significant increase by the DCV doses of subjects who received from 0 doses (28.57%) to 5 doses (79.17%) (*p* < 0.001) (Table 1). The positivity rate of tetanus antibody for women of childbearing age was 7.52% (95% CI: 3.08, 14.90), significantly lower than overall subjects.

In the multivariable regression adjusted analyses with a stepwise method, the interval since the last vaccination, sex, and area were significantly associated with diphtheria toxoid IgG-specific antibody levels; age, area, and doses were significantly associated with tetanus toxoid IgG-specific antibody levels (Table 2).

The percentage of subjects with diphtheria or tetanus toxoid antibody levels differed from age groups (*p* < 0.001). The percentage of those with protection (≥0.1 IU/mL) in children younger than 15 was much higher than those ≥ 15 years old (Figure 2 and Figure 3).

## 4. Discussion

The results of community-based serosurveillance indicated that diphtheria toxoid antibody concentration and positivity rate were decreased by age and increased by doses, as were tetanus toxoid antibody concentration and positivity rate. The tetanus toxoid IgG-specific antibody levels and positivity rate for women of childbearing age were significantly lower than overall subjects. In the multivariable models, area and doses were significantly associated with diphtheria and tetanus toxoid antibody concentration.

In concordance with prior epidemiological studies, a decline in the antibody level of diphtheria or tetanus was observed. We found that immunity to diphtheria and tetanus waned with age and was lowest in older subjects. A seroepidemiology of diphtheria study in Beijing, China, in 2012 showed that the positivity rate was highest in children aged 1–4 y (97.63%) and lowest in subjects over 40 y (34.11%) [23]. A serosurveillance study in patients from 2018–2019 in Chongqing, China, showed that the proportion of diphtheria positive levels decreased from 76.9% among children aged 18 m–3 y to 29.0% among those aged 40–50 years [3]. Wang found that the diphtheria antibody positivity rate dropped from 96.75% for children aged 0–9 years old to 7.14% for people over 40 years old, the tetanus antibody positivity rate dropped from 96.75% for children aged 0–9 years old to 7.14% for people over 40 years old [24]. A meta-analysis showed that the tetanus positivity rate decreased along with growing older, from 88.9% for children aged 0–7 to 52.9% for people aged 16–40 [25].

The diphtheria antibody levels waned after vaccination. Grasse found that 24% of the young and 100% of the elderly could not provide long-lasting (5 years) protection after booster vaccinations against diphtheria [26]. There is neither a diphtheria/tetanus-containing vaccine booster during pregnancy nor booster vaccinations against diphtheria/tetanus used in teenagers and adults in China, except for five doses of diphtheria/tetanus-containing vaccine for children, as mentioned before. Around the world, boosters for DT were administered to teenagers, adults, and pregnant women to boost their antibodies. According to the World Health Organization Immunization data (accessed on 7 December 2023), 117 countries or regions arranged a diphtheria booster dose for pregnant women, 130 countries or regions arranged a diphtheria booster dose for adolescents between 9–18 years old, 93 countries or regions arranged 1–5 diphtheria booster doses for adults; 126 countries or regions arranged a tetanus booster dose for pregnant women, 130 countries or regions arranged a tetanus booster dose for adolescents between 9–18 years old, 99 countries or regions arranged 1–5 tetanus booster doses for adults.

In recent years, several older patients carrying non-pathogenic, non-toxigenic corynebacterium diphtheria have been reported in or near Guangdong, Guangzhou [4,5,6]. The monitoring data in China indicated that most of the tetanus patients were over 40 years old, with a relatively high case fatality rate [3]. Compared to children and adolescents, middle-aged people and the aged are at much higher risk of infection with Corynebacterium diphtheriae and Clostridium tetani. No DTC booster dose is administrated after six years old, the diphtheria and tetanus antibody levels decline over time, and the diphtheria and tetanus antibody for adults and women of childbearing age was relatively low. According to the changes in the diphtheria or tetanus situation, the current Chinese diphtheria or tetanus immunization schedule does not meet the requirements. An urgent DT booster dose is needed for middle-aged people, women of childbearing age, and the elderly in China. However, there are no DT dosage forms for adults in China; a DT dosage form for adults urgently needs development.

The diphtheria or tetanus antibody levels have no statistical significance between local people and foreigners, as mentioned in a previous study [27]. Most adults and children have been immunized since China launched the planned regular vaccination program with DTaP in 1978. The result reflected that the planned regular vaccination program was successfully and fairly evenly implemented in different regions in China.

Previous domestic and foreign serosurvey studies had shown a moderate positivity rate against diphtheria/tetanus, like our study (61.82%/71.61%), from 55.6% to 66.28% [23,28,29,30]/50.47% to 78.60% [25,27,31,32,33,34]. Some research chose the cut-off above 0.01 IU/mL as the minimum protective level for diphtheria and tetanus [32,35]. They believed that the threshold of ≥0.10 IU/mL was likely to underestimate the levels of protection because studies in human and animal models have shown that 0.01 IU/mL was a protective level of immunity for diphtheria/tetanus [36,37]. In the WHO position paper, antibody levels above 0.1 IU/mL confer complete protection [38,39]. For safety considerations, the higher cut-off value (0.1 IU/mL) was chosen in this study. However, cases of tetanus have been documented in individuals with antibody concentrations above these thresholds [38].

Research indicates that humoral immune responses to immunization and infection and susceptibilities to antibody-mediated autoimmunity are generally lower in males [40]. However, this study drew the opposite conclusion: the male sex was associated with having higher antibody levels and a higher rate of seropositivity for both diphtheria and tetanus in the univariable analysis, though the sex difference only persisted for diphtheria in the multivariable analysis. Further analysis found that vaccine coverage of 4 or more doses of DTaP was higher in males (77.73% vs. 84.31%, Appendix A). However, the effect of gender persists for diphtheria after adjusting for number of vaccines received. Perhaps there is some residual confounding related to unmeasured vaccination-related gender differences, e.g., differential timing of vaccination by gender. Studies in China [24,34,41], and Tohme’ study in Nigeria, 2018 [42] found the same result. These findings raise the question of whether gender-based inequity in access to vaccination services in Guangzhou may exist. More specific studies may be needed to fully understand the relationship between vaccine coverage and seropositivity by gender, and gender-based inequity in access to vaccination services.

Diphtheria and tetanus antibody positivity rates were higher in both central urban and suburban area than in outer suburban area. The difference was partially related to underlying vaccination rates among areas. The constituent ratio of booster DTaP doses (4, 5 doses) was found to be lower in the Outer suburban area than in the other two areas (82.78% (Central urban) vs. 86.21% (Suburban) vs. 74.30% (Outer suburban), (Appendix A). The convenience and accessibility to vaccination services in central urban and suburban areas are superior to outer suburban areas, which may lead to a higher booster DTaP vaccination rate in central urban areas.

The declining trends in routine immunization coverage can lead to accelerated falling antibody levels. On 15 July 2018, it was revealed that a total of 0.50 million substandard DTaP vaccines produced by Changsheng Biotechnology were sold in Shandong and Anhui, and 400,520 substandard DTaP vaccines produced by the Wuhan Institute of Biological Products were sold in Hebei and Chongqing. On 25 July, the National Medical Products Administration in China clarified that the affected vaccine would be ineffective, but receiving the ineffective vaccines would not result in harmful health consequences [43]. Although there have been no reports of illnesses or deaths related to the faulty vaccines to date, the scandal has caused panic among Chinese parents. Respondents from seven cities not involved in the scandal reported a significant decrease in confidence in domestically-produced vaccines [44]. Studies after the vaccine crisis showed that parents were hesitant or refused to vaccinate their children, and childhood immunization rates diminished in a short period [21].

Fortunately, the government took steps to re-establish confidence in vaccines: launched an investigation into all vaccine producers across China, punished 357 officials over the vaccine scandal, recalled the vaccines in question, and started vaccination campaigns aimed at providing supplementary vaccines. To improve the vaccine management system and re-establish the confidence of Chinese people in domestic vaccines, the Standing Committee of the National People’s Congress, China’s highest legislature, adopted the first comprehensive national law on vaccine management, The Vaccine Administration Law [45].

The current study displays a serologic estimate of DTaP under real-world conditions among the general health population. The results from this seroepidemiological study in younger children may provide some evidence to evaluate the DTaP’s effectiveness. It is likely to be a valuable means of allaying the public’s panic and restoring confidence in domestic vaccines.

We acknowledge that the sample size and composition are limitations of our study. The subjects are primarily children, and adult and older age groups are underrepresented. People over 14 and 60 only account for 18.74% and 0.84%, respectively, significantly lower than Guangzhou’s actual popularity composition proportion. This might affect the reliability of the results among adults and the elderly; the results of adults and the elderly may not be repeated, and these conclusions cannot be generalized to the older age groups. Improving the design of the surveillance of immunity levels and increasing the sample size of adults and older adults as appropriate is necessary. Continuous and well-designed surveillance is needed to monitor the change in diphtheria or tetanus antibody levels in adults and older people.

Second, given the year of development of the information system and the risk of recall bias, we failed to collect the vaccination records of a fraction of children (5.51%) and people over 14. Adults are too old to record in the system, and the children may be the transient population living in Guangzhou; if they do not receive vaccines in Guangzhou, they are unlikely to extract their vaccination files from other cities, resulting in no vaccination records in the system. Missing vaccination records may be more frequent among unvaccinated people, which can result in a positive bias to the results.

Third, because the subjects’ occupation information was not collected in the study, the antibody level for post-traumatic tetanus high-risk occupation cannot be analyzed.

## 5. Conclusions

The present study revealed a decline in diphtheria or tetanus antibody levels in people over 14. The current diphtheria and tetanus immunization schedule does not provide persistent protection after childhood. There is an urgent need to adjust the current diphtheria and tetanus immunization schedule to fill this immunity gap. According to the WHO position paper [38,39], at least one booster dose of the diphtheria vaccine and tetanus vaccine should be given in combination at 9–15 years of age to maintain a protective antibody in adults; pregnant women should have at least two doses of tetanus toxoid -containing vaccines, preferably DT, with an interval of at least four weeks between doses and the second dose at least two weeks before the birth, to ensure pregnant women and their newborn infants are protected from birth-associated tetanus. Constant actions are also needed to boost the immunization rate.

## Figures and Tables

**Figure 1 vaccines-12-00381-f001:**
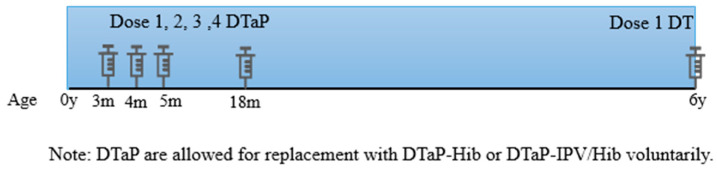
China′s latest diphtheria and tetanus immunization schedule.

**Figure 2 vaccines-12-00381-f002:**
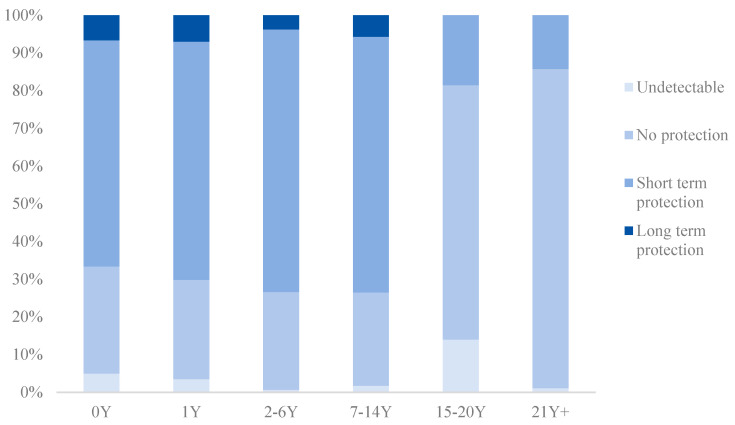
Percentage of subjects with diphtheria toxoid antibody levels from different age groups.

**Figure 3 vaccines-12-00381-f003:**
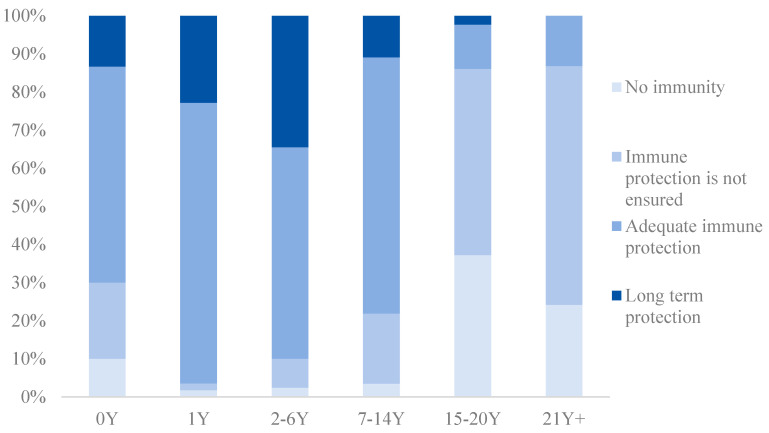
Percentage of subjects with tetanus toxoid antibody levels from different age groups.

**Table 1 vaccines-12-00381-t001:** The Diphtheria and Tetanus toxoid IgG-specific antibody levels for healthy population in 2018.

		Diphtheria		Tetanus	
Group	No. (%)	GMC (IU/mL, 95%CI)	Positivity Rate (≥0.1 IU/mL, %, 95%CI)	GMC (IU/mL, 95%CI)	Positivity Rate (≥0.1 IU/mL, %, 95%CI)
**Age**					
0–1 y	117 (16.36)	0.151 (0.119, 0.191)	68.37 (59.13, 76.66)	0.319 (0.238, 0.428)	84.35 (76.39, 90.45)
0 y	60 (8.39)	0.125 (0.086, 0.180)	66.67 (53.31, 78.31)	0.169 (0.103, 0.277)	70.00 (58.06, 81.94)
1 y	57 (7.97)	0.183 (0.136, 0.247)	70.18 (56.6, 81.57)	0.623 (0.513, 0.757)	96.49 (91.57, 101.42)
2–6 y	290 (40.56)	0.165 (0.147, 0.186)	73.45 (67.97, 78.44)	0.476 (0.412, 0.551)	90.00 (86.53, 93.47)
7–14 y	174 (24.34)	0.161 (0.137, 0.189)	73.56 (66.36, 79.95)	0.230 (0.189, 0.28)	78.16 (71.96, 84.36)
15–20 y	43 (6.01)	0.032 (0.023, 0.045)	18.60 (8.39, 33.4)	0.017 (0.01, 0.029)	13.95 (3.16, 24.74)
21 y-	91 (12.73)	0.048 (0.041, 0.056)	14.29 (7.83, 23.19)	0.023 (0.018, 0.031)	13.19 (6.10, 20.27)
**Sex**					
Male	354 (49.51)	0.161 (0.142, 0.183)	71.75 (66.75, 76.38)	0.298 (0.25, 0.356)	80.51 (76.36, 84.66)
Female	361 (50.49)	0.099 (0.088, 0.111)	52.08 (46.79, 57.33)	0.149 (0.124, 0.18)	62.88 (57.87, 67.89)
**Household registration**					
Local	421 (58.88)	0.135 (0.014, 1.323)	74.35 (69.89, 78.45)	0.253 (0.027, 2.323)	74.35 (70.16, 78.54)
Nonlocal	291 (40.7)	0.133 (0.119, 0.149)	67.35 (61.64, 72.71)	0.224 (0.19, 0.263)	67.35 (61.93, 72.77)
**Area**					
Central urban	235 (32.87)	0.198 (0.168, 0.234)	66.38 (59.95, 72.39)	0.318 (0.256, 0.396)	77.45 (71.56, 82.63)
Suburban	240 (33.57)	0.127 (0.114, 0.141)	67.5 (71.56, 82.63)	0.259 (0.218, 0.308)	79.58 (73.92, 84.5)
Outer suburban	240 (33.57)	0.081 (0.07, 0.094)	51.67 (45.15, 58.14)	0.115 (0.089, 0.15)	57.92 (51.4, 64.24)
**Doses**					
0	7 (0.98)	0.040 (0.007, 0.219)	28.57 (−16.56, 73.7)	0.015 (0.003, 0.068)	14.28 (3.16, 58.87)
1	9 (1.26)	0.096 (0.044, 0.21)	55.56 (15.04, 96.07)	0.194 (0.048, 0.778)	66.67 (29.93, 92.51)
2	8 (1.12)	0.062 (0.015, 0.253)	50.00 (5.31, 94.69)	0.114 (0.019, 0.684)	62.50 (24.49, 91.48)
3	82 (11.47)	0.133 (0.101, 0.176)	65.85 (55.37, 76.34)	0.284 (0.202, 0.398)	82.92 (73.02, 90.33)
4	317 (44.34)	0.164 (0.146, 0.184)	72.87 (67.95, 77.79)	0.440 (0.381, 0.508)	88.64 (84.63, 91.92)
5	144 (20.14)	0.193 (0.164, 0.228)	79.17 (72.45, 85.88)	0.282 (0.226, 0.353)	84.72 (77.79, 90.17)
**Interval since last vaccination (Y)**					
1	172 (24.06)	0.21 (0.177, 0.249)	79.89 (73.15, 85.57)	0.489 (0.401, 0.596)	91.38 (86.18, 95.09)
2	112 (15.66)	0.18 (0.153, 0.212)	76.79 (67.86, 84.24)	0.485 (0.406, 0.578)	94.64 (88.7, 98.01)
3	100 (13.99)	0.164 (0.137, 0.197)	76.24 (66.74, 84.14)	0.361 (0.277, 0.47)	87.13 (79, 92.96)
4	68 (9.51)	0.121 (0.093, 0.158)	60.87 (48.37, 72.4)	0.313 (0.22, 0.446)	81.16 (69.94, 89.57)
5	39 (5.45)	0.142 (0.106, 0.192)	69.23 (52.43, 82.98)	0.241 (0.165, 0.351)	82.05 (66.47, 92.46)
6	43 (6.01)	0.086 (0.055, 0.134)	58.14 (42.13, 72.99)	0.113 (0.067, 0.192)	62.79 (46.73, 77.02)
Total	715 (100)	0.126 (0.115, 0.137)	61.82 (58.14, 65.39)	0.210 (0.185, 0.240)	71.61 (68.3, 74.92)

**Table 2 vaccines-12-00381-t002:** Epidemiologic Correlates the Diphtheria and Tetanus toxoid IgG-specific antibody levels for healthy population in 2018.

	Multivariable Models			
	Diphtheria		Tetanus	
Group	β (95% CI)	*p*	β (95% CI)	*p*
Age	-		−0.123 (−0.161, −0.084)	<0.001
Interval	−0.005 (−0.006, −0.003)	<0.001	-	
Sex			-	
Female	Ref.		-	
Male	0.119 (0.048, 0.19)	<0.05	-	
Area				
Central urban	Ref.		Ref.	
Suburb	−0.295 (−0.383, −0.208)	<0.001	−0.246 (−0.372, −0.12)	<0.01
Outer suburb	−0.344 (−0.436, −0.252)	<0.001	−0.339 (−0.475, −0.203)	<0.001
Doses				
0	Ref.		Ref.	
1	-		-	
2	−0.32 (−0.731, 0.091)	>0.05	−0.268 (−0.854, 0.318)	>0.05
3	−0.018 (−0.323, 0.286)	>0.05	−0.047 (−0.479, 0.386)	>0.05
4	0.116 (−0.18, 0.412)	>0.05	0.494 (0.051, 0.938)	<0.05
5	0.263 (−0.037, 0.564)	>0.05	0.946 (0.429, 1.463)	<0.01

## Data Availability

Data are unavailable due to privacy or ethical restrictions.

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
