# Peer review of "Seroprevalence of Diphtheria and Tetanus Immunoglobulin G among the General Health Population in Guangzhou, China"

_vaccines, 2024, doi:10.3390/vaccines12040381_

Round 1

Reviewer 1 Report (Previous Reviewer 1)

Comments and Suggestions for Authors

The manuscript has been significantly improved after the round of revisions.

Author Response

We appreciate your summary of the manuscript and encouraging comment.

Reviewer 2 Report (Previous Reviewer 2)

Comments and Suggestions for Authors

 I appreciate the author’s careful attention and thoughtful responses to my questions. I have a few remaining clarifications and comments below, which are all quite minor.

Previous major comments:

1.       Previous major comment #1: I appreciate the author’s discussion of the gender differences in immunization coverage and their potential role in the observed gender differences in seropositivity. The call for further research into gender-based inequities in vaccination coverage is helpful to highlight as well.

In the newly added paragraph, some light proofreading and clarification would be helpful. For instance, there should probably be a colon after “conclusion” in line 326. In line 329-331, there is a run-on sentence, which should probably read something like “Studies in China [23,30,40] and a study by Tohme et al in Nigeria in 2018 [41] found the same result. These findings raise the question of whether there may be gender-based inequity in access to vaccination services in Guangzhou.” I am not sure what the constituent ratio mentioned in line 331 is – is this simply vaccine coverage of 4 or more doses of DTaP? And it would be helpful to clarify whether this is further analysis of data that the authors have used throughout the rest of the manuscript, if these coverage values come from a different data source [if the latter, would need a citation].

Last, I would also note that the authors’ multivariable models do attempt to adjust for number of doses of vaccine delivered, and that the diphtheria gender difference persists even after adjusting for number of doses of vaccine. In theory, if the gender difference in antibody levels is due to differences in vaccine coverage, the effect of gender should no longer be present after adjusting for number of vaccines received?  Perhaps there is some residual confounding related to unmeasured vaccination-related gender differences, e.g. differential timing of vaccination by gender, etc.? In any case, I would suggest that the authors might want to add a comment to this discussion explaining why they think that the difference may have persisted despite adjustment for number of doses delivered in the multivariable analysis. (Or, at least, noting that it does persist for diphtheria after adjustment for # of doses delivered, and stating that more specific studies may be needed to fully understand the relationship between vaccine coverage and seropositivity by gender in this population).

2.       Previous major comment #2: I also appreciate the added discussion by the authors re: differences in antibody levels by urbanicity. The added information about vaccination rates is helpful also. I have the same general comments as above:

a.       Can the authors clarify in the text whether the vaccination coverage rates are from the authors’ own data or from a different source, and cite any external sources? 

b.       The authors may want to soften the language in line 336 to say that the difference may be *partially* due to different underlying vaccination rates between these areas. The authors find that antibody levels are higher in central urban areas than suburban and outer suburban areas. The vaccine coverage data, however, has a slightly different pattern – with only outer suburban areas’ coverage being lower than central urban areas, but suburban areas actually having higher coverage than central urban areas. So this may partially explain the observed findings with respect to outer suburban areas, but it’s harder to use these coverage patterns to explain why suburban areas would have lower antibody levels (than central urban areas) if their coverage is actually higher)

c.       More broadly, in this new paragraph, the authors hypothesize that the differences in antibody levels may be related to differences in vaccination coverage and in age distributions between the different areas – that is, confounding by age and confounding by vaccination status. But the authors included both age and number of doses in the stepwise covariate selection process for the multivariable regression analysis, which – in theory – should adjust for these factors. (Age is included only in the tetanus model, but doses are included in both the diphtheria and tetanus final models). Why do the authors think that the differences between central urban, suburban, and outer suburban areas persist even after adjustment? Is this due to residual confounding, for instance?

Previous minor comments

·         Previous minor comment #3: Abstract: I still see some formatting issues in a couple of the estimates in the abstract (line 16, 61.82%(95CI%:58.14,65.39) and 71.61%(95CI%:68.3,74.92)). These should be formatted as (95% CI: XX, XX) (with spaces for consistency with the rest of the paper and for easier reading.

·         Previous minor comment #5: Thanks to the authors for adding citations and would continue to defer to the editors on journal standards for handling of unpublished data (line 48).

·         Previous minor comment #9, Re: “multivariable regression analysis with a stepwise  method”. Thanks to the authors for the clarifications. I would suggest adding two points of detail to improve reproducibility, specifically how the “most contributive predictors” were determined, and how “improvement in model fit” was determined (e.g. AIC, etc.): “The strategy of stepwise regression is stepwise selection (or sequential replacement), which started with no predictors, then sequentially add the most contributive predictors [based upon _____]. After adding each new variable, remove any variables that no longer provide an improvement in the model fit [as measured by _____].”

·         Previous minor comment # 10: Thanks for re-writing the inclusion and exclusion criteria; defer to editors as to whether the current formatting is consistent with journal style or whether they would prefer these to be written as sentences (e.g. “we included subjects who were 1) without fever… etc. and excluded outpatients and hospital inpatients).” Also, I would clarify what “outpatients” means for the exclusion criteria – outpatients can include anyone who is receiving medical care for any condition outside of the hospital setting, so theoretically this could include (for instance) an adult who is receiving blood pressure treatment or a child who is being treated for an unrelated disorder like eczema. Were all individuals receiving any medical care for any condition excluded from the study?

·         Previous minor comment #12: I appreciate the reformatting of Table 1 which is very helpful. Just as an aside, the “Household registration” and “interval since last vaccination”  headers in table 1 are now indented farther than other headers, but I expect that this would be taken care of in proofing.

·         Previous minor comment #14, Re: non-neonatal tetanus” – I appreciate the clarification. One note: “is no a national notifiable disease in China” should be “is not a national notifiable disease in China”

·         The remaining comments are all well -addressed by the authors; no further questions on these.

Comments on the Quality of English Language

I have tried to point out a few additional areas above where some grammatical edits or clarifications would be helpful, but the authors have addressed many of these already. I have especially aimed to point out areas where clarification is needed to help the reader understand the content, but there are very few of these issues in the current draft.  I would defer to the editors for further guidance - in general, my impression is that a thorough proofread would be helpful prior to publication just to ensure that any remaining typos or grammatical inconsistencies are fixed. 

Author Response

Thank you very much for taking the time to review this manuscript. A point-by-point response to comments and suggestions is shown in the attachment. Language has been polished.

Reviewer 3 Report (Previous Reviewer 3)

Comments and Suggestions for Authors

The authors have addressed the questions raised in the original submission. English can still be improved. 

Comments on the Quality of English Language

Quality of the English language can still be improved. 

Author Response

Thank you for your constructive comments regarding our paper. Language has been polished.

This manuscript is a resubmission of an earlier submission. The following is a list of the peer review reports and author responses from that submission.

Round 1

Reviewer 1 Report

Comments and Suggestions for Authors

The authors propose a seroprevalence study aiming to assess the impact on immunity levels against tetanus and diphtheria in the population after the distribution of batches of vaccines with manufacturing defects in 2018, rendering those vaccines ineffective from an immunological standpoint.

I believe the study design may not effectively address this question, both due to the sample selection and the lack of vaccination records. However, as a seroprevalence study, it may have some local interest in the context of evaluating and planning vaccination campaigns in Guangzhou. The results obtained do not provide significant new insights into the already-known aspects of the topic and are unlikely to be generalizable.

However, as a seroprevalence study, it may have some local interest in the context of evaluating and planning vaccination campaigns in Guangzhou. The results obtained do not provide significant new insights into the already-known aspects of the topic and are unlikely to be generalizable.

Some recommendations for the authors:

1. Please provide more detailed explanations of the vaccine registration system from which certain data are extracted.

2. Please clarify the difference between DCV dose records and DCV vaccination date.

3. Please correct typographical errors:

    - Line 97: "Comparrpe" → "compare".

   - Line 152: The acronym DCV appears for the first time without having been explained before.

    - Line 203: In Table 1, the table appears disordered.

    - Line 208: In Table 1, is it a second table different from the previous one? It would need to be renumbered.

These suggestions aim to enhance the clarity and precision of the study, as well as address potential errors in the text.

Following the recommendations of the STROBE Statement, I would suggest improving the following sections:

 - In the introduction, the role attributed to the vaccine distribution episode in relation to the objectives and hypothesis is not clear. In my opinion, it would be more appropriate to use it in the discussion as an element of explanation or understanding of the obtained results.

 - Enhance the definition of the objectives and the underlying hypothesis.

 - Improve the explanation of eligibility criteria and the various sources of information. It would be advisable to provide more details about the sources and quality of data obtained from health records.

 - Analyze/describe potential sources of bias due to sample effects. There is a clear underrepresentation of older age groups.

 - Specify how missing data in the affected variables have been handled.

Reviewer 2 Report

Comments and Suggestions for Authors

Overview: In this manuscript, the authors report on a seroprevalence survey in Guangzhou, China in 2018, assessing antibody levels against diphtheria toxoid and tetanus toxoid. The survey was conducted using a multistage stratified sampling approach, and results were analyzed both in terms of the geometric mean concentration and using categorical groupings to proxy seroprotection. The authors find that antibody levels (and seroprotection categorizations) drop precipitously after adolescence and differ by both geographic area, number of doses, and sex. In the discussion, they correlate these findings to a lack of teen / adult booster doses or vaccination during pregnancy in the current vaccine schedule in China (and a lack of available Td vaccine formulations for adults), calling for a need to adjust the current immunization schedules to fill this immunity gap.

Overall, the manuscript is easy to follow (though could benefit from some light proofreading and copy-editing), the serosurvey and analysis appear to have been well-conducted, and the findings are of substantial public health importance. I have some questions below for clarification, but these are mostly minor.

Major comments:

1. The authors find that male sex was associated with having higher antibody levels and a higher rate of seropositivity for both diphtheria and tetanus in the univariable analysis (though the sex difference only persisted for diphtheria in the multivariable analysis). I don’t see this result discussed in the discussion – do the authors have any theories as to why there might be a difference in seropositivity by sex? Is this related to underlying vaccination rates?

2. Similarly, the analyses suggest that central urban areas have higher seropositivity rates and antibody levels for both diphtheria and tetanus toxoids, compared to suburban or outer suburban areas. It would be useful for the authors to comment on this in the discussion – do they have any theories about what might be driving these differences and/or the implications for vaccination service delivery strategies? 

Minor comments:

1. Abstract: I think that (95CI%:XX,XX) should be (95% CI: XX, XX) (with spaces, etc.). 

2. I am not familiar with the term corynephalous and a brief internet search doesn’t reveal any definitions – is this a typographical error? (line 38)

3. Line 46: Data from National Health Commission needs a citation; similarly, the unpublished data in line 49 needs some sort of attribution. Defer to editors on the journal standards for handling of unpublished data.

4. Line 90-91: the parenthetical citation (CFDA, 2018) should probably be numbered and refer to the reference list, like the other citations.

5. Methods section: it would be helpful for the subheadings (participants, serologic evaluations, etc.) to have some other formatting (underline, italics) and/or to be offset from the text with spaces for easier reading. Defer to editors on journal standards for formatting.  

6. Methods section: did the authors’  lab protocols include any internal and/or external quality control on the assays?  If so, would suggest that those processes be included in the methods.

7. Methods section, lines 153-154. The authors describe a “multivariate regression analysis with a stepwise  method”

a. I think that this should be *multivariable*? (It is later described as multivariable in line 205-206, which seems more appropriate

b. Could the authors give a little more detail about the stepwise method (presumably this is a stepwise method for variable selection, but is this forward selection, backward elimination, etc.?) Just another sentence or half-sentence with some clarification would help with reproducibility of the analytic methods described.

8. The inclusion criteria (lines 122-126) are written as a list, but it would be helpful for these to be written as complete sentences to fit with the rest of the manuscript. 

9. Lines 136-144: the breakpoints for diphtheria are given as <0.01,  0.01 - < 0.1, 0.1 – 1.0, and > 1.0. For tetanus, however, they are given as < 0.01, 0.01 - < 0.1, **0.11** – 1.0, and > **1.1**. Shouldn’t these also be given as 0.1 – 1.0 and > 1.0 for consistency (and for complete coverage of the potential range of values, i.e. 1.05)? Also, the 0.01 - <0.1 IU/mL range is defined as “no protection” for diphtheria but as “protection not ensured” for tetanus toxin in the text, but it appears that in figures 1 and 2, “not ensured” is used for both. It would be helpful to be consistent with the labeling.

10. Table 1: it would be helpful to reformat the columns so that the parentheses don’t break onto the next page, for easier reading (or so that the 95% Cis in parentheses are all on the next line). I would also suggest making the subheaders (Household Registration, Sex, Doses, Age, etc.) bold or somehow otherwise more visually apparent). 

11. The authors refer to the “new China vaccine administration law” in lines 78-79 and 286-287. I would suggest that the authors provide a bit more context / description of the law for readers who may be unfamiliar with it. 

12. Line 303: "WHO position paper" needs a citation

Comments on the Quality of English Language

As above, the manuscript is quite easy to follow. I would suggest some light copyediting and proofreading. I've flagged some of this in the minor comments above. For instance:

- Line 49: "no neonatal tetanus is no a national notifiable disease" - should this read "although neonatal tetanus is a national notifiable disease" (since the rest of the sentence talks about a lack of surveillance amongst non-neonatal patients)?

- Line 62: "vaccine ... were" should be "vaccines... were" or "vaccine... was"

- Lines 74-76: "The level of antibodies against infectious diseases measured by serosurveillance resulted from vaccination or previous infection history, which was an indicator of the immune ability of individuals to diseases and the effectiveness of the target vaccine in the vaccinated population." - this should probably be written in the present tense, e.g. as "The level of antibodies against infectious diseases, as measured by serosurveillance,  result from vaccination or previous infection history and can be used as an indicator of the immune ability of individuals to [respond to] diseases and the effectiveness of the target vaccine in the vaccinated population." 

- Line 95: "according to the actual situations" - should this be "to assess the current situation" or something similar? 

- Line 97: I think that there is a typographical error ("comparrpe" shoudl be "compare")

- Line 112: "community stuffs" should probably be "community staff"?

- Line 117: "Considering the information system developed year and subjects’ recall bias" could be re-written for clarity, e.g. as "given the year of development of the information system and the risk of recall bias, we only collected"... 

- Throughout, I would probably call it the "positivity rate" rather than "positive rate" (line 229 uses "positivity rate", which I think is more standard)

- There are a few parts of the discussion that are somewhat conversational, e.g. line 258 "pretty low".

- Line 257-259 needs to be split into two sentences or somehow otherwise combined, as it is currently a run-on sentence: "The tetanus antibody for women of 257 childbearing age was pretty low, it might be the possible reasons for neonatal tetanus still 258 reported in China."

- Line 283: "steps to re-establishing" should be "steps to re-establish"

- Line 284: Officials shouldn't be capitalized

- Line 303-307: "According to the WHO position paper, at least one booster dose of the diphtheria vaccine and tetanus vaccine should be given in combination at 9–15 years of age to maintain a positive antibody in adult, and pregnant women should have a certain tetanus vaccination history to ensure pregnant women and their newborn infants are protected from birth-associated tetanus." Here, "to maintain a positive antibody in adult" should probably say "to maintain seropositivity in adults" (or more importantly, to maintain protective immunity rather than just seropositivity). Also, "a certain tetanus vaccination history" is somewhat vague, and it would be helpful to be more specific".

- Line 307-308: "Constantly" should probably be "constant" or "ongoing".

Author Response

Thank you very much for taking the time to review this manuscript. Please find the detailed responses below and the corresponding revisions/corrections highlighted in the re-submitted files.

Reviewer 3 Report

Comments and Suggestions for Authors

General Comments: this is a serological survey assessing the prevalence of protective antibodies against two toxins: diphtheria and tetanus toxins. It is a very important topic and crucial for assessing the effectiveness of vaccination programs and evaluating the level of herd immunity to tetanus and diphtheria diseases in the population. It should be of interest to the readership of the journal.

Specific Comments:

1.       Samples and Sampling: Sampling was done as part of an annual monitoring program in Guangzhou. Although the procedure for multi-stage stratified sampling was detailed, along with an excellent description of sample size calculations, the authors did not indicate the estimated population size from which they are sampling 442 samples. This information is important for the reader to appreciate this section better.

2.       Community stuffs (line 112)? What is that? The authors might want to change or explain what this is, just to avoid confusion.

3.       Guangdong Vaccine Circulation and Vaccination Management Information System: The authors need to explain the reliability of this database. The phrase: “considering the information system developed year and subjects’ recall bias…” (line 117 – 118) is very confusing. Are the authors saying that the data obtained here were from surveys? Was information entered into the system by physicians at the time of vaccination? Are the data subjective or objective? If the authors are going to make conclusions about length of time from vaccination, the conclusions must be based on reliable data.

4.       The schedule of vaccination in China ensures that the primary series is concluded at 2 years of age. How many two-year-olds have antibodies? How many of them have protective levels of antibodies? This study should be able to answer these questions by changing table 1 so that participants are grouped as follows:

·         0 – 2years

·         2 – 6 years (booster doses are not given until age 6. This would be a group that allows the authors to comment about rate of antibody decline after completing primary vaccination series.

·         7 – 12 years (booster doses are given at 12?). Again, this group should allow the authors to comment about declining antibody levels following the 6-year booster.

How many people received booster doses at 6 and 12 years old? These are important information to improve this manuscript.

5.       Local resident vs migrant residents: These two groups of participants in this study were not defined in the methods section. Is this something related to length of stay in the community?

6.       Line 161: DCV vaccination records: The authors answered some of the questions in #3 above by showing that 549 participants had vaccination records for DCV. Are these records the same as tetanus records?

7.       Line 177. Are the authors still talking about antibodies to tetanus toxoid here, or diphtheria toxoid?

Comments on the Quality of English Language

Good. 

Author Response

(The authors gave the same response as above.)
